

# TaRTLEt: Transcriptionally-active Riboswitch Tracer Leveraging Edge deTection

Sachit Kshatriya and Sarah C. Bagby

Department of Biology, Case Western Reserve University, Cleveland, OH, United States of America

## ABSTRACT

Structured RNAs have emerged as a major component of cellular regulatory systems, but their mechanism of action is often poorly understood. Riboswitches are structured RNAs that allosterically regulate gene expression through any of several different mechanisms. *In vitro* approaches to characterizing this mechanism are costly, low-throughput, and must be repeated for each individual riboswitch locus of interest. Bioinformatic methods promise higher throughput; despite robust computational identification of riboswitches, however, computational classification of the riboswitch mechanism has so far been both model-bound, relying on identification of sequence motifs known to be required for specific models of riboswitch activity, and empirically untested, with predictions far outpacing biological validation. Here, we introduce TaRTLEt (Transcriptionally-active Riboswitch Tracer Leveraging Edge deTection), a new high-throughput tool that recovers *in vivo* patterns of riboswitch-mediated transcription termination from paired-end RNA-seq data using edge detection methods. TaRTLEt successfully extracts transcription termination signals despite numerous sources of biological and technical noise. We tested the effectiveness of TaRTLEt on riboswitches identified from a wide range of sequenced bacterial taxa by utilizing publicly available paired-end RNA-seq readsets, finding broad agreement with previously published *in vitro* characterization results. In addition, we use TaRTLEt to infer the *in vivo* regulatory mechanism of uncharacterized riboswitch loci from existing public data. TaRTLEt is available on GitHub and can be applied to paired-end RNA-seq datasets from isolates or complex communities.

## INTRODUCTION

Riboswitches are highly conserved, widespread gene regulatory elements that exploit RNA's potential for complex structure to build ligand-responsive differential regulation of gene expression directly into an mRNA. Encoded in the untranslated regions (UTRs) of genes, riboswitches are highly specific aptamers for a wide variety of small-molecule ligands. In the two decades since the discovery of the first riboswitch (*Mironov et al., 2002*), we have learned that riboswitches are compact, specific, and responsive to transient changes; that they often regulate critical organism functions; and that they operate *via* diverse and complex modes (*Mandal et al., 2003*; *Amadei et al., 2023*; *Sudarsan et al.,*

Corresponding author
Sarah C. Bagby, scb126@case.edu

*2005*). Given these properties, riboswitches have been investigated for uses ranging from synthetic sensors (*Wachsmuth et al., 2013*; *Boussebayle et al., 2019*), to logic elements in complex gene expression systems (*Hanson et al., 2003*; *Groher & Suess, 2014*), to potential targets for antimicrobials and novel therapeutics (*Blount & Breaker, 2006*; *Ellinger et al., 2023*; *Giarimoglou et al., 2022*). Yet our understanding of natural riboswitch function remains patchy, with high-confidence computational identification of candidate riboswitch sequences and ligands (*Nawrocki, 2014*; *Nawrocki & Eddy, 2013a*; *Chang et al., 2009*; *Eddy & Durbin, 1994*; *Yao et al., 2007*; *Stav et al., 2019*) far outpacing our understanding of their activity *in vivo*.

This gap is widened by the breadth of riboswitches' regulatory repertoire. Biochemical methods such as transcript fragment length and ribosome binding assays (*Winkler et al., 2003*; *Nou & Kadner, 2000*; *Welz & Breaker, 2007*; *Hollands et al., 2012*) have identified an overall pattern of riboswitch regulation: riboswitch-mediated allosteric regulation of expression begins with selective binding of a ligand to the non-coding aptamer domain, stabilizing the domain's conformation (*Gilbert et al., 2006*) and altering the structure of the expression platform. But this broad pattern includes riboswitches that affect any of several different levels of gene expression. Most straightforwardly, riboswitch ligand binding can sequester or expose conventional expression control logic elements like intrinsic terminators/antiterminators (*Mironov et al., 2002*; *Mandal & Breaker, 2004*) or Rho-binding sites (*Hollands et al., 2012*) to alter mRNA production, or translation start sites to alter protein synthesis (*Breaker, 2018*). However, riboswitch mechanisms can also go beyond interaction with direct control architectures. Thiamine pyrophosphate (TPP) riboswitches in the fungus *Neurospora crassa* alter gene expression by controlling mRNA splicing (*Cheah et al., 2007*); the *glmS* riboswitch-ribozyme in *Bacillus subtilis* recruits exonucleases for transcript degradation by catalyzing upstream mRNA cleavage (*Collins et al., 2007*; *Klein & Ferré-D'Amaré, 2006*); the *Escherichia coli lysC* riboswitch both sequesters the Shine-Dalgarno (SD) sequence and exposes an RNase E binding site (*Caron et al., 2012*); and a SAM riboswitch in *Listeria monocytogenes* is capable of acting in *trans* on a distal target (*Loh et al., 2009*), among numerous other examples documented to date (*Bédard, Hien & Lafontaine, 2020*; *Ariza-Mateos, Nuthanakanti & Serganov, 2021*).

To understand the role riboswitches play in a given microbe's physiology, we need to understand at what regulatory level each acts. Regulatory mechanisms have been established for at least one member of the riboswitch families binding roughly a dozen ligands, so that in principle we might expect to infer the mechanism used by other members of the same families. But even in the small set of families with ≥2 biochemically characterized members, we find cases where different members of a family act by different mechanisms (*Barrick & Breaker, 2007*). Thus, robust identification of riboswitch ligands is insufficient to predict regulatory mechanism even for comparatively well-studied riboswitch families, while for many more families, no representative riboswitch has yet been characterized *in vitro*. *In silico* methods seeking to fill this gap have largely relied on DNA-level signals, using the computation of folding energies (*Gong et al., 2017*) coupled with the identification of SD sequences, U-rich terminator motifs, or Rho-binding sites to determine states that could alter transcription/translation efficiency (*Barrick & Breaker, 2007*; *Sun & Rodionov,*

*2014*). While these analytical methods can produce *a priori* predictions about riboswitch regulatory modes, they leave unexplored a more direct readout of the realized activity of riboswitches within living cells: the distribution of fragments captured in RNA-seq data.

Here, we describe the new tool TaRTLEt (Transcriptionally-active Riboswitch Tracer Leveraging Edge deTection), which applies high-throughput computational approaches to RNA-seq data to determine which of the known riboswitches identified in a (meta)omic dataset show evidence of regulating gene expression by altering transcription termination efficiency. We hypothesized that different experimental conditions would, directly or indirectly, evoke changes in riboswitch ligand concentrations that might alter riboswitch regulatory state. Under this hypothesis, we predicted that, when the range of conditions tested included both above- and below-threshold ligand concentrations, transcription-attenuating riboswitches would produce a distinctive coverage pattern at riboswitch loci in paired-end RNA-seq datasets; and, further, that this pattern could be robustly identified using computational approaches developed for edge detection (*Canny, 1986*). Importantly, this pattern should support distinguishing riboswitch-mediated transcription termination from the broader set of changes in gene expression that could be evoked by other transcriptional regulators (*e.g.*, transcription factor proteins) acting at these loci. We show that the occurrence of this pattern at riboswitch loci identified in publicly available transcriptomic datasets is in good agreement with existing *in vitro* characterization of riboswitch regulatory modes. By extracting information from the traces of riboswitch activity that are left in the RNA pool of cells actively regulating gene expression, our approach complements and extends earlier methods based on inference from DNA sequence. Our high-throughput approach can identify signatures of riboswitch-mediated transcription termination in existing or new (meta)transcriptomic data, greatly expanding not only the set of riboswitch loci with data-driven predictions of regulatory mode but also our understanding of the range of growth conditions under which each transcription-terminating riboswitch is active.

## MATERIALS & METHODS

Portions of this text were previously published as part of a preprint (https://doi.org/10.1101/2024.10.15.618519).

### Theoretical framework
#### *Predicted transcript pools for transcriptionally active riboswitches*
Riboswitches are often associated with transcription start sites (TSS), allowing transcription start site analysis to serve as a strategy for identifying novel riboswitches and regulatory RNA (*Rosinski-Chupin, Soutourina & Martin-Verstraete, 2014*; *Rosinski-Chupin et al., 2019*; *Adams et al., 2017*; *Yu, Vogel & Förstner, 2018*). We reasoned that a transcription-terminating riboswitch in the 5′ UTR of a gene should produce different patterns of RNA-seq read coverage in the ON *vs.* OFF state (Figs. 1A, 1B). In either case, transcription should begin at the TSS and proceed through the riboswitch itself. In the OFF state, formation of the transcription-terminator structure should truncate transcripts near the riboswitch 3′ end (Fig. 1A top). By contrast, in the ON state, RNA polymerase should

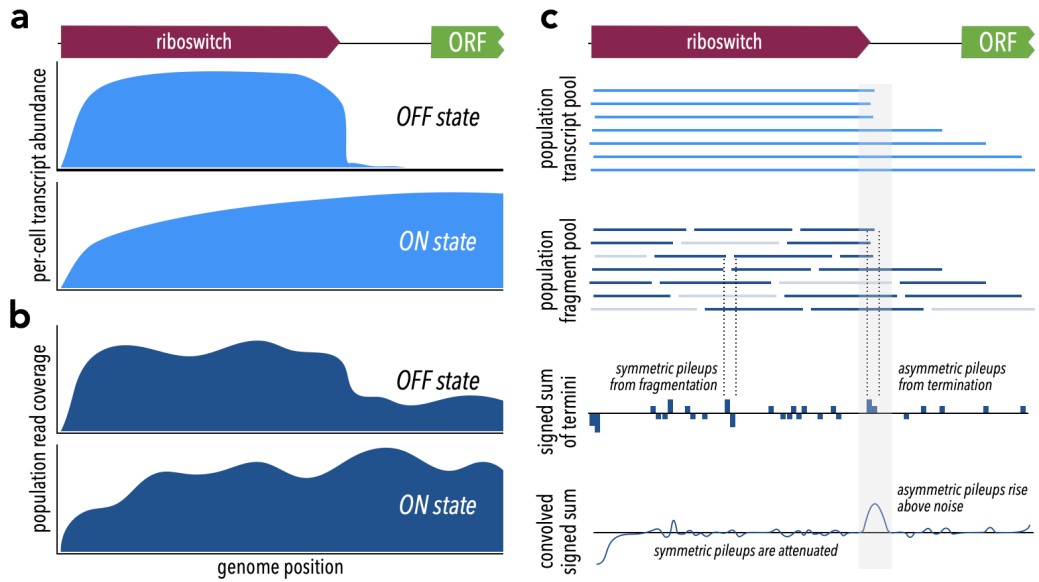

**Figure 1** **Theoretical framework for TaRTLEt's approach.** (A) At riboswitches that act by modulating transcription termination, a single cell's transcript abundance profile differs sharply between the riboswitch OFF and ON states. (B) When RNA is extracted from a population of cells for sequencing, coverage profiles are made noisier by a combination of biological (*e.g.*, subpopulations in ON states alongside others in OFF states) and technological factors (*e.g.*, transcript fragmentation, sequencing of a subset of the fragment pool). (C) Weighted convolution on the signed sum of fragment termini attenuates the noise signal, making transcription termination signals readily detectable even when the population-level read coverage does not fall to zero.

read through, producing transcripts that extend well into the downstream gene (Fig. 1A bottom). A riboswitch that produces these two patterns under different conditions exhibits behavior consistent with riboswitch-mediated conditional regulation of transcription termination.

### Fragment distributions and terminal pileups

What relation does the coverage map of sequencing reads bear to the pool of transcripts? Each RNA-seq read represents some region of a transcript, but because transcripts are typically fragmented prior to sequencing we know only that the 5′ end of a read must align with a fragment terminus and not whether that fragment is internal or terminal to a transcript. Fragmentation often, as in the validation dataset used here, results in a random biased distribution (Fig. S1) of fragment sizes. Importantly, however, the new 3′ and 5′ termini produced by each fragmentation event come in neighboring pairs. Even though the total number of fragment termini, and thus the total number of read termini, may far exceed the number of original transcript termini, we can still extract a transcript size signal by considering pileups of read termini. Because paired-end sequencing generates reads that represent both ends of each sequenced fragment, the resulting data offers a higher signal:noise ratio than single-end sequencing: there, fragmentation is more extensive *and* at most one terminus of each fragment is represented in the read set, so that comparable sequencing depth yields greater stochastic fluctuations in the set of termini captured. We

therefore designed `TaRTLEt` v1.0 to restrict analysis to paired-end datasets. Future versions could be extended to analyze appropriately deep single-end datasets.

We reasoned that a convolution approach would allow us to both extract the needed transcript size signal and smooth the overall signal. Convolution is heavily used in applications like edge-detection algorithms (*Canny, 1986*; *Albawi, Mohammed & Al-Zawi, 2017*), where it can act as an attenuating filter for neighboring signals. Here, we want neighboring pileups of 3′ and 5′ read termini to drop out, filtering out transcript-internal fragmentation to reveal the transcript-terminal signal. Tracking 3′ ends as +1 and 5′ ends as −1, we can first compute the signed sum of read termini for each position in the reference (Fig. 1C), then calculate a weighted average of surrounding sums for each position. We call the resulting convolution's features "peaks", and we seek to identify peaks at which a pileup of 3′ termini is not accompanied by a pileup of 5′ termini immediately downstream.

The choice of kernel function for computation of the weighted average affects the filter's selectivity for changes in coverage. Rather than a flat uniform distribution, we chose to use kernels constructed by discretizing a Gaussian distribution, such that tuning the spread of this distribution (by the standard deviation $\sigma$) tunes the filter's selectivity for how sharply coverage must change. Because sites of transcription termination, especially intrinsic termination, are constrained (*Roberts, 2019*; *Gusarov & Nudler, 1999*), riboswitch-mediated transcription termination should produce a sharp drop in coverage. To match this expectation, we might use a tight convolution (small $\sigma$) kernel that attenuates peaks arising from changes over >1–2 nt. However, both biological and technical factors (*e.g.*, thermodynamic fluctuations; processes in library preparation, sequencing, and quality control) are expected to broaden the biological signal to some degree. Through trial and error, we chose a default $\sigma$ of 1.5 for kernel generation, corresponding to a filter that passes coverage changes localized within a 6-nt region.

Regardless of the number of fragmentation events, the convolution always returns non-zero peaks at positions with asymmetric pileups of 5′ or 3′ termini. These surviving peaks should represent a range of events: transcription termination should be relatively rare and should cause a large, tightly centered coverage drop, giving rise to high, narrow peaks; sporadic processes (*e.g.*, artifacts in library prep or sequencing) should be relatively common and yield lower, broader peaks. While these differences in peak shapes should provide an opportunity to further refine the set of candidate peaks, the probability and details of the processes generating background peaks may be sequence-dependent in as yet poorly understood ways, complicating comparison to a global baseline. Instead, we reasoned that we could collect a per-locus "noise set" of peaks within a defined neighborhood scaled to riboswitch size. Fitting a two-dimensional Gaussian noise distribution to this collection of peaks provides a basis for comparison, allowing us to robustly identify unusually sharp peaks at each locus as candidate sites of riboswitch-mediated transcription termination.

### Probing fragment-end distributions across treatments

For any of these sites to reveal conditional regulation by a transcription-terminating riboswitch, the efficiency of termination (that is, the proportion of transcripts that terminate) near the riboswitch 3′ end must vary across treatments that evoke ON and

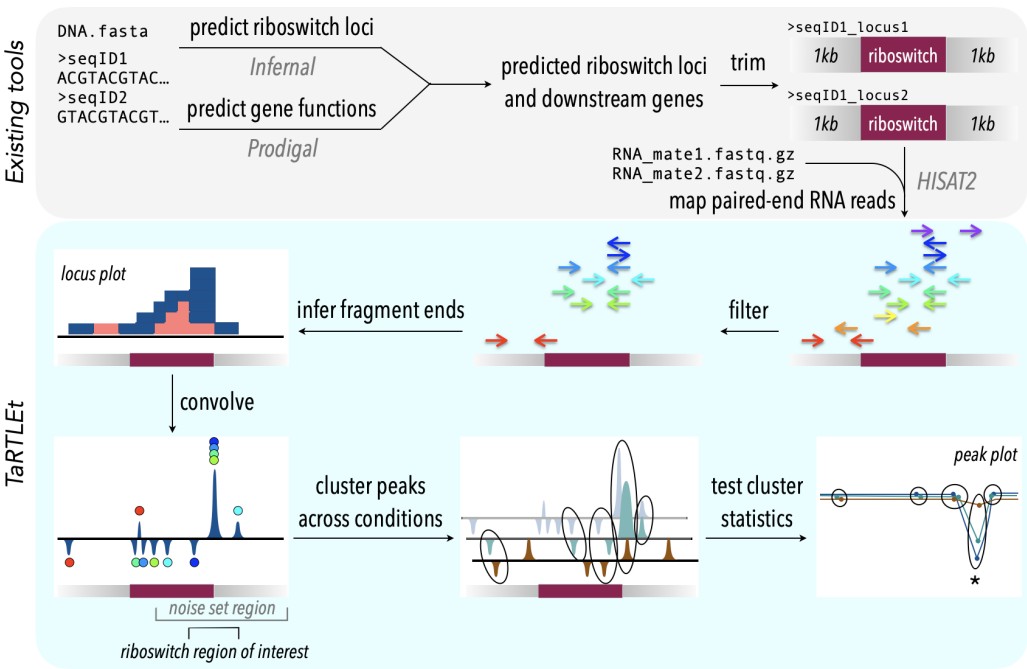

**Figure 2 TaRTLEt pipeline for detection of transcription-terminating riboswitches in paired-end RNA-seq data.** Existing tools (top) are used to identify known riboswitch families in (meta)genomic data, predict downstream open reading frames, and map RNA-seq reads to a reduced dataset of riboswitch loci and their downstream regions. TaRTLEt then filters mapped reads by orientation and analyzes the distribution of fragment termini, generating a signed sum of termini at each position and applying a weighted convolution to attenuate the symmetric signals arising from mRNA fragmentation before sequencing. Peaks in the convolution are clustered across conditions by genome position; then TaRTLEt analyzes changes in fractional read coverage across each cluster to identify sites of condition-dependent transcription termination. Locus plots are generated for each riboswitch locus under each experimental condition represented in the RNA-seq data, to support per-locus per-condition calling of riboswitch ON or OFF state; peak plots capture all experimental conditions, to support per-locus calling of riboswitch regulatory mode.

OFF states. For a riboswitch that acts by transcription termination, termination efficiency should be low (low or no coverage drop) in the ON state and high (large coverage drop) in the OFF state. We bound the region of interest for this search to a range of $0.5 \times$ riboswitch size both upstream and downstream of the riboswitch $3'$ end and examine the fractional coverage change (that is, (change in coverage across peak)/(average coverage depth across entire riboswitch)) at each convolution peak in the region under each treatment condition. To compare results at coincident peaks across treatment conditions, we account for the possibility that peak location might shift slightly due to small variations in process noise by clustering peak locations across treatment conditions (Fig. 2). Any cluster that captures both ON and OFF states of a transcription-terminating riboswitch should exhibit fractional coverage change with a mean that is significantly more negative and a variance that is significantly larger than baseline (Figs. 2 and 3).

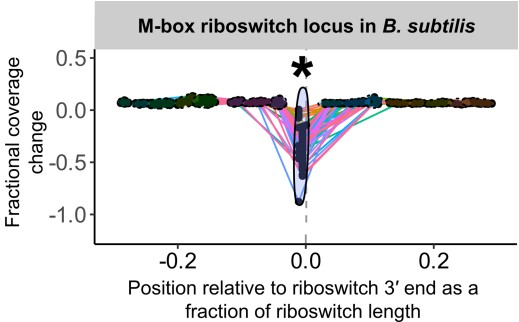

**Figure 3** **Sample peak plot output of `TaRTLEt`, showing evidence of conditional transcription termination at a riboswitch 3′ end.** For each riboswitch locus, the locus plot arising from each experimental condition is convolved and fragment terminus peaks are detected. Each point shows the fractional coverage change across one such peak. Colored lines connect peaks detected under the same experimental conditions. Peak locations are shown relative to riboswitch size and 3′ end; that is, a peak at 0.1 on the *x*-axis lies 10% of the riboswitch length downstream of the riboswitch 3′ end. Peaks are clustered across experimental conditions based on x-position; ellipses mark peak clusters. All detected peaks are clustered, but only clusters containing at least one transcription-termination peak are viable candidates. Significance testing of the mean and variance of each such cluster's fractional coverage drop identifies clusters consistent with conditional transcription termination by the riboswitch locus (asterisk).

## `TaRTLEt` implementation

`TaRTLEt` is implemented in `python` v3.11 with a dedicated command-line interface (CLI). Its usage does not require knowledge of `python` and the included `conda` environment environment contains installations for any external CLI tools. For widespread utility, we chose to incorporate a riboswitch identification pipeline into our approach, so that inputs may be unannotated genomes/sequences; identified riboswitch loci are then submitted to the mechanistic inference pipeline. A complete walk-through of `TaRTLEt` usage and the analysis of data generated by `TaRTLEt` is available at the tool GitHub repository (https://github.com/lepton-7/tartlet). The pipeline is summarized conceptually in Fig. 2 and in more detail in Fig. S2.

### *Identification of riboswitches and open reading frames*

Input sequences in FASTA format are processed through `Infernal` v1.1.5 (http://eddylab.org/infernal) (*Nawrocki & Eddy, 2013b*) and `Prodigal` v2.6.3 (https://github.com/hyattpd/Prodigal) (*Hyatt et al., 2010*). For `Infernal`, each input sequence is passed to `cmscan` using the `--cut_ga --rfam --nohmmonly --noali --tblout` options against a comprehensive riboswitch covariance model of families compiled from the latest Rfam collection (version 14.9 as of writing) (https://rfam.org/) (*Griffiths-Jones et al., 2003*). `cmscan` output is parsed to identify riboswitch loci for further processing. `Prodigal` is called with the default output and translation options. For each identified riboswitch, the downstream gene is identified from the `Prodigal` output and recorded for downstream plot annotations. The downstream gene is defined as the first open reading frame past the riboswitch 5′ end and on the same strand. This definition captures open reading frames

(ORFs) that overlap with part of the identified riboswitch sequence, in order to maximize the likelihood of capturing ORFs that are biologically relevant to the riboswitch.

### Riboswitch reference generation

To reduce computational requirements for transcript mapping, TaRTLEt trims input genomic sequences back to riboswitch neighborhoods, producing reference sequences that stretch from 1,000 nt upstream of each identified riboswitch locus to 1,000 nt downstream. (For metagenomic inputs, where riboswitch loci may lie less than 1,000 nt from contig ends, neighborhoods are permitted to be smaller or asymmetric.) We chose this neighborhood size on the basis of common practice in paired-end sequencing library preparation, where the target fragmentation size is often less than 1,000 nt to preserve base quality (*Tan et al., 2019*). Consequently, the generated reference neighborhoods are typically (96.61% of riboswitch-aligned reads in our validation dataset) large enough to capture both a riboswitch-aligned read and its paired-end mate. All references generated by TaRTLEt are oriented 5′→ 3′; references derived from (-) strand riboswitches are reverse complemented before saving to file.

### Reference alignment

RNA-seq reads are aligned to the generated riboswitch reference using HISAT2 (http://daehwankimlab.github.io/hisat2/) (*Kim et al., 2019*). HISAT2 was chosen for its fast per-read alignment time, good parallel scaling, and acceptable alignment rates (*Musich, Cadle-Davidson & Osier, 2021*). HISAT2 calls are made using the default options, but any additional valid alignment options may be passed through the TaRTLEt interface. The current TaRTLEt implementation requires that the input RNA-seq datasets be paired-end. The resulting SAM files are converted to sorted BAMs for downstream processing.

We recommend running HISAT2 using the options `--no-unal --score-min L,0,−0.4`. The use of `--no-unal` dramatically reduces output file sizes by omitting unaligned reads. The use of `--score-min L,0,−0.4` decreases stringency to allow reads with alignment mismatches/soft-clipping to be included in the output; any alignment errors are handled during downstream processing. The validation dataset presented here was run using these parameters in addition to `-p 40` (performance tuning with multiple threads) and `-t` (wall-time logging).

### Processing mapped reads

TaRTLEt heavily utilizes the `pysam` (https://github.com/pysam-developers/pysam) library (a python interface for `samtools` (*Li et al., 2009*) and `htslib` (*Bonfield et al., 2021*)) to parse BAM files and generate alignment data objects for downstream data transformation. Once mapped reads/read pairs have been collected from the sorted BAMs, they are filtered to discard pairs in invalid orientations (RF, TANDEM, FFR, RRF), which may reflect sequencing errors. Only pairs in the EQ and FR orientations, and pairs with only one mapped read, are retained as valid. The EQ ("equal") orientation denotes reads aligned in opposite directions that overlap across their entire length, as is often the case when the fragment sequenced is ∼150 bp, comparable in scale to individual short reads. In the FR orientation, the left-most aligned base (looking 5′→ 3′ in the reference) is either

shared between the F and R oriented reads or belongs only to the F oriented read, and the right-most aligned base belongs exclusively to the R read.

Next, TaRTLEt generates fragment coverage pileups using the filtered set of reads for every position in the reference sequence. Single reads (with unmapped mates) are discarded during counting unless the --allow-single-reads flag is passed to TaRTLEt. TaRTLEt tracks five types of coverage. Two can incorporate data from single reads if included: "read coverage" counts bases directly mapped by a read, while "clipped fragment coverage" counts bases mapped by HISAT2's soft-clipped read regions. The remaining three types require both mates to be mapped. "Overlapped fragment coverage" counts bases overlapped by both mates; "inferred fragment coverage" counts unmapped bases that lie between the mapped regions of the two mates; and "fragment termini coverage" counts only the mapped base that is the origin of read synthesis, *i.e.,* looking $5' \rightarrow 3'$ in the reference, the left-most mapped base for an F read and right-most mapped base for an R read. This last category captures the ends of sequenced fragments to feed into the signed sum of termini for convolution analysis; we count the fragment terminus captured by the F read as $-1$ and the fragment terminus captured by the R read as $+1$. Reads that have soft-clipped regions are discarded from this step unless the --allow-soft-clips flag is specified. Fragment termini counts are removed from the read coverage and/or overlapped coverage counts, and read coverage counts are removed from the overlapped coverage counts, such that no mapped base is counted more than once. This ensures that counts across all coverage types at a given position sum to the read depth at that position.

Finally, to ensure sufficient coverage for further analysis, riboswitch loci are filtered based on the average read coverage across the length of the riboswitch. The coverage threshold can be adjusted with the --min-cov-depth option; the validation presented here uses the default value of 15.

### Transformation and feature identification

Fragment termini coverage pileups are transformed using a convolution to extract information about transcription starts and stops. TaRTLEt generates a 51-element one-dimensional weighted kernel by discretizing a Gaussian probability density function with $\sigma = 1.5$. The weighted kernel is then applied to the fragment termini coverage array through the convolution to generate an array of peaks corresponding to transcription events. The convolution transforms coverage at each position into convolution amplitude (arbitrary units).

TaRTLEt conducts simple peak detection as follows for a positive peak: a local convolution amplitude maximum is defined as the peak "summit". The peak region covers all the bases on either side of the summit at which convolution amplitude (a) is monotonically decreasing and (b) does not enter or cross a zero-region, defined as the interval $(-0.1, 0.1)$. To delineate ambiguous shoulder regions between peaks, peak bounds follow pythonic inclusive-exclusive convention where the left peak bound represents the left-most valid base according to the monotonicity and zero-crossing criteria and the right peak bound represents the first base that fails the criteria. Negative- and positive-amplitude peaks are considered candidate transcription start and termination events, respectively.

Next, TaRTLEt assesses candidate transcription-termination peaks individually to determine which rise above background. Working with per-locus, per-condition convolution plots, TaRTLEt compares each positive peak within a user-defined region to a two-dimensional multivariate normal (MVN) distribution fitted to the widths and amplitudes of each other peak in the region (the "noise set"). The region that feeds into the noise set is defined relative to riboswitch length, with parameters defining the proportion of the riboswitch that should be excluded at the 5′ end and how far past the riboswitch 3′ end the region should stretch. (Our validation analysis used `--ext-prop -0.3 1.0` to define a noise set region that excludes the first 30% of the riboswitch, includes the remainder, and continues beyond the riboswitch 3′ end to include a region equal in length to the complete riboswitch.) The likelihood of drawing a given peak from the MVN is used as a pseudo *p*-value, with a significance threshold of $< 0.05$. Peaks that pass this test and exceed a user-configurable coverage change threshold (by default, a drop of $\geq 20\%$ of the mean coverage across the locus) are called as transcription termination events.

To facilitate biological interpretation of riboswitch activity, TaRTLEt then asks, for each condition and each locus, whether any passing transcription-termination peaks were observed. This within-condition check does not in itself offer evidence for riboswitch regulation by transcription termination. Instead, it (1) feeds into the cluster-level calls described below; (2) supports manual curation of results; and (3) allows the user to ask, among the transcription-terminating riboswitches identified below, what set of experimental conditions evoke the ON state *vs.* the OFF state.

### Peak clustering and statistics

To detect differential transcription termination, TaRTLEt compares data from all treatment conditions at each riboswitch locus. All peaks that fall within the riboswitch region of interest (user-configurable; by default, within $\pm 0.5 \times$ riboswitch size of the riboswitch 3′ end) under any condition are clustered by position to enable cross-condition comparisons. The riboswitch 3′ end defines position 0, and peak positions are given relative to this site as a fraction of the riboswitch size. For example, a position of $-0.15$ would indicate that the peak is located 15% of the riboswitch size upstream of the riboswitch 3′ end. Clustering is implemented *via* `fclusterdata` from the `SciPy` package and follows a cophenetic distancing algorithm with complete linkage. The default (user-configurable) cophenetic distance threshold of 0.04 was chosen through trial and error (Fig. S3).

Once peaks are assigned to clusters, cross-cluster comparisons at each locus are used to test for condition-dependent transcription termination. Each peak is characterized by the change in total fragment coverage across its bounds. The fractional coverage change is defined as the difference in total coverage between the left and right bounds of the peak, expressed as a fraction of the average total read coverage between the 5′ and 3′ ends of the riboswitch. TaRTLEt performs a Mann–Whitney one-tailed U-test (`scipy.stats.mannwhitneyu`) to compare each cluster's fractional coverage change to the mean fractional coverage change of all other peaks, to test whether the cluster mean is more negative than the mean of the set of all other peaks.

Testing for differences in cluster variance must account for bias from differences in cluster sizes. TaRTLEt uses Levene's test (`scipy.stats.levene`) to run pairwise comparisons between the variance of a given cluster enclosing $n$ peaks and that of random samples of $n$ peaks chosen without replacement from all other peaks at the same riboswitch locus until the superset of peaks is exhausted or at least 60 comparisons have been made. If the superset is exhausted before 60 comparisons are made, all peaks are replaced and the clusters are randomly resampled to reach 60 comparisons. TaRTLEt returns the median of the resulting 60 $p$-values for each peak cluster.

Finally, TaRTLEt combines peak- and cluster-level tests to assess significance. To be called as showing evidence of condition-dependent transcription termination, a riboswitch locus must have at least one cluster that (i) includes at least one significant transcription-termination peak (*i.e.,* distinct from the noise-set MVN at $p < 0.05$ *and* coverage drop greater than or equal to the user-defined threshold; see previous section); (ii) passes the Mann–Whitney U-test for mean fractional coverage change; and (iii) passes the Levene test for increased variance.

### Description of outputs

TaRTLEt outputs data on two layers of abstraction: single-condition convolution peaks, and cross-condition peak clusters. At each level, TaRTLEt generates both tabular output on each feature and plots of each riboswitch locus. For each candidate peak in the convolution, the single-condition tabular output (`peak_log.csv`) captures the riboswitch locus, the RNA-seq dataset, the location and shape of the peak, the "noise set" of comparator peaks, TaRTLEt's pass/fail call for that peak as a candidate transcription termination event, and the reason for that call. Single-condition locus plots are sorted into `pass/` and `fail/` subdirectories, where `pass/` contains locus plots for all conditions in which any peak, regardless of location within the locus, was called as a transcription termination event. A sample locus plot of the molybdenum cofactor (MOCO) riboswitch from *E. coli* is shown in Fig. 4; for a sample peak log, see Table S1.

At the level of cross-condition cluster comparisons, TaRTLEt records each cluster's mean and variance test statistics in `cluster_stats.csv`, then uses this file in conjunction with `peak_log.csv` to generate a summary peak plot that allows easy inspection of condition-dependent modulation of transcription termination efficiency near the 3′ ends of all riboswitches in the dataset. Figure 5 shows a sample summary peak plot. The user may simply scan the summary peak plot to identify loci in which a cluster was marked significant (*), optionally using the locus plots to identify particular experimental conditions that evoked a change in riboswitch transcription termination efficiency (at loci identified as modulating transcription termination) or to spot-check peak call quality.

### Validation datasets

All the paired-end RNA-seq datasets used to test and characterize TaRTLEt here are presented in Table 1. These datasets were obtained from the Gene Expression Omnibus (GEO) (*Barrett et al., 2012*). We used the Ribocentre-switch database (*Bu et al., 2024*) to facilitate identification of previously characterized riboswitches in this dataset.

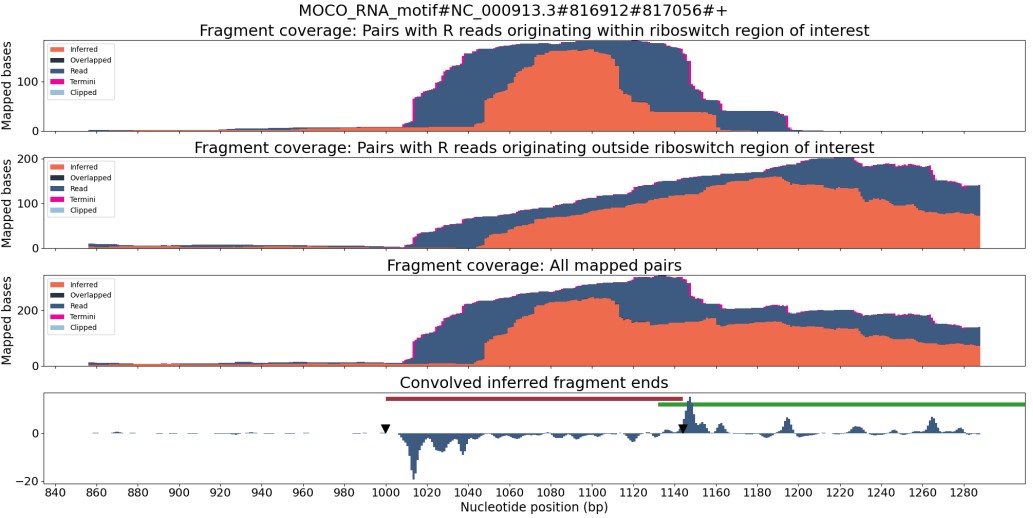

**Figure 4** **Locus plot of the MOCO riboswitch from *E. coli* for RNA-seq sample SRR7154624.** The top panel collects fragment coverage for read pairs with R-oriented reads starting in the riboswitch region of interest ($\pm 0.5 \times$ riboswitch size). The second panel collects fragment coverage for read pairs with R-oriented reads starting outside the region of interest. Coverage counts in the third panel are the union of the top two panels. The bottom panel records the results from applying the Gaussian convolution to the fragment termini coverage. Vertical axes for all panels are not normalized and are scaled to fit the maximum displayed value.

# RESULTS

## TaRTLEt performs well on known transcriptionally-active riboswitches

To validate this new tool, we ran TaRTLEt on publicly available paired-end RNA-seq datasets for 31 microbial genomes (Table 1). Of these, datasets for 29 genomes were successfully processed through TaRTLEt. This collection of microbes is phylogenetically diverse, with good representation of both Gram-positive and Gram-negative bacteria (Fig. 6A), and includes 21 riboswitch loci that have previously been experimentally characterized.

While this low count underscores the need for high-throughput methods to supplement *in vitro* approaches, it also presents a practical impediment to robust determination of error rates. In light of this limitation, TaRTLEt uses conservative analytical and statistical approaches and thresholds, as well as making intermediate analysis steps available to the user for inspection. Our 29-genome validation dataset included 405 riboswitch loci and a total of 655 RNA-seq readsets. Manual curation of TaRTLEt's locus plots led us to exclude one genome (*P. fluorescens*, 10 loci, 40 readsets) as poorly mapped. Among the remaining 28 genomes and 395 loci, TaRTLEt identified 115,130 coverage-change features as potential transcription-termination peaks; comparison to transcriptomic noise allowed us to reject all but 3,437 candidates. In cross-condition significance testing of peak clusters, 1,765 of these candidates lay within the 104 clusters at 95 riboswitch loci that passed our mean and variance test criteria. TaRTLEt calls riboswitch loci as showing evidence of condition-dependent transcription termination based on this latter, most conservative,

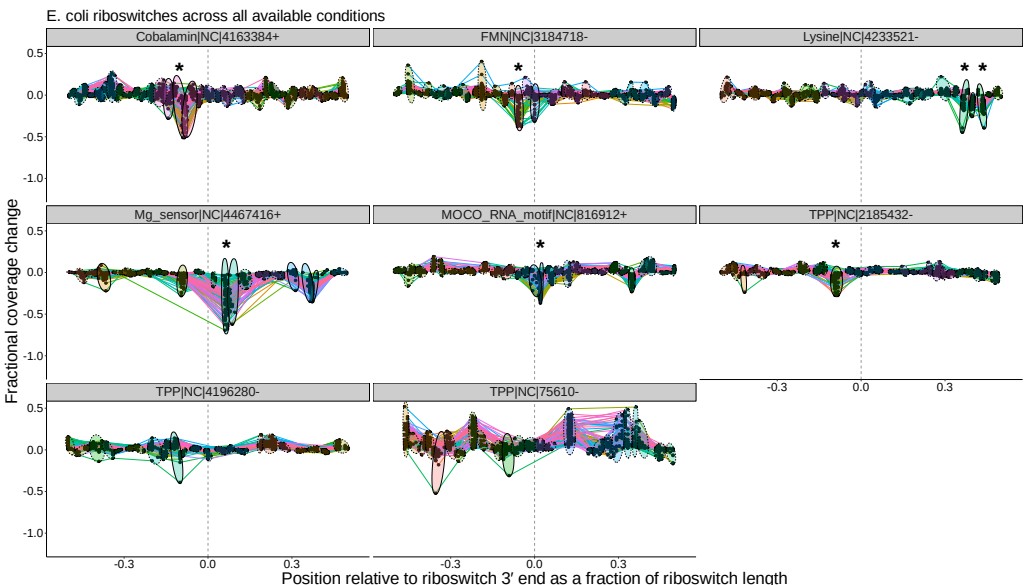

**Figure 5** **Peak plots (see Fig. 3) for the eight *E. coli* riboswitch loci identified by `Infernal` in the *E. coli* genome.** Peak clusters are grouped by colored ellipses; clusters with at least one significant transcription-termination peak are bounded by solid ellipses, and those with no significant peaks are bounded by dashed ellipses. Asterisks (*) mark peak clusters that (i) contain at least one significant transcription-termination peak, (ii) have significantly lower mean ($p < 0.05$, Mann–Whitney one-tailed U-test) than the set of all other peak clusters, and (iii) have significantly higher variance ($p < 0.05$, Levene's test) when compared to randomly sampled sets of the same number of peaks.

most data-rich set, while the within-condition single-peak calls (and supporting data) are made available to the user to support manual curation.

Overall, our results are in good agreement with literature findings (Fig. 6B). `TaRTLEt`'s peak plots (Fig. 5, Figs. S4, S7) showed clear condition-dependent transcription termination at 10 of the 19 loci known to use this regulatory mode: in *B. anthracis*, the upstream tandem TPP riboswitch (*Welz & Breaker, 2007*); in *B. subtilis*, the *lysC* lysine, *mgtE* M-box, *xpt* purine, and *samT* and *metK* SAM riboswitches (*Sudarsan et al., 2003*; *Dann et al., 2007*; *Mandal & Breaker, 2004*; *Winkler et al., 2003*); and in *E. coli* the *btuB* cobalamin, *ribB* FMN, *lysC* lysine, and *thiM* TPP riboswitches (*Nou & Kadner, 1998*; *Hollands et al., 2012*; *Sedlyarova et al., 2016*; *Bastet et al., 2017*). In addition, `TaRTLEt` detected no indication (Figs. S4, S6) of transcription-termination activity at either of the two negative-control loci experimentally characterized as acting strictly through mechanisms other than transcription termination (the *B. subtilis* *glmS* and *Enterococcus faecalis* $S_{MK}$ riboswitches (*Collins et al., 2007*; *Fuchs, Grundy & Henkin, 2007*; *Smith et al., 2010*)).

We also examined the validation dataset's performance across the pipeline to identify the range of situations that can give rise to inconclusive or negative results, as observed at the nine remaining loci in our dataset with previous experimental characterization. First, `TaRTLEt` cannot provide insight into riboswitch loci with read coverage too low to support statistical analysis. We observed this outcome in 29 of the 395 riboswitch loci investigated

**Table 1  List of GEO accessions for paired-end RNA-seq read sets used for characterizing riboswitches from each organism.**

| Microbe | GEO accession | Citation |
| --- | --- | --- |
| *Acinetobacter baumannii* | GSE183334 | Unpublished |
| *Aliivibrio fischeri* ES114 | GSE237189 | *Vander Griend et al. (2024)* |
| *Apilactobacillus kunkeei* | GSE205998 | *Seeger et al. (2023)* |
| *Bacillus anthracis* | GSE152356 | *Corsi et al. (2021)* |
| *Bacillus subtilis* str. 168 | GSE219221 | *Guo & Herman (2023)* |
| *Bacillus subtilis* str. 168 | GSE226559 | Unpublished |
| *Bacteroides fragilis* | GSE220692 | *Fiebig et al. (2024)* |
| *Bacteroides thetaiotaomicron* | GSE129572 | *Briliūtė et al. (2019)* |
| *Bacteroides thetaiotaomicron* | GSE169260 | *Lewis & Gui (2023)* |
| *Bacteroides xylanisolvens* | GSE74379 | *Despres et al. (2016)* |
| *Burkholderia pseudomallei* | GSE152295 | *Avican et al. (2021)* |
| *Caulobacter vibrioides* | GSE241057 | *McLaughlin, Fiebig & Crosson (2023)* |
| *Clostridioides difficile* | GSE173804 | *Weiss et al. (2021)* |
| *Desulfovibrio vulgaris* str. Hildenborough | GSE101911 | *Chen et al. (2019)* |
| *Desulfovibrio vulgaris* str. Hildenborough | GSE78834 | *Gao et al. (2016)* |
| *Enterococcus faecalis* | GSE152295 | *Avican et al. (2021)* |
| *Escherichia coli* | GSE114358 | *Guzmán et al. (2019)* |
| *Escherichia coli* | GSE122211 | *Sastry et al. (2019)* |
| *Escherichia coli* | GSE122295 | *Sastry et al. (2019)* |
| *Escherichia coli* | GSE122296 | *Sastry et al. (2019)* |
| *Escherichia coli* | GSE122779 | *Anand et al. (2019)* |
| *Eubacterium limosum* | GSE149269 | *Jeong et al. (2020)* |
| *Klebsiella pneumoniae* | GSE229867 | *Liu et al. (2025)* |
| *Mycobacterium tuberculosis* str. H37Rv | GSE218354 | *Martini et al. (2023)* |
| *Mycolicibacterium smegmatis* MC2 155 | GSE232901 | *Grigorov et al. (2023)* |
| *Neisseria gonorrhoeae* | GSE191020 | *Ray et al. (2022)* |
| *Piscirickettsia salmonis* | GSE235725 | *Carril et al. (2023)* |
| *Pseudomonas chlororaphis* subsp. Aureofaciens 30-84 | GSE61200 | *Wang et al. (2016)* |
| *Pseudomonas fluorescens* | GSE200822 | *Hasnain et al. (2023)* |
| *Salmonella enterica sv.* Typhimurium | GSE203342 | *Kant et al. (2023)* |
| *Sinorhizobium meliloti* | GSE151705 | *Fagorzi et al. (2021)* |
| *Staphylococcus epidermidis* | GSE152295 | *Avican et al. (2021)* |
| *Streptococcus sanguinis* SK36 | GSE174672 | *Puccio et al. (2022)* |
| *Streptomyces coelicolor* | GSE234439 | *Li et al. (2023)* |
| *Synechococcus elongatus* PCC7942 | GSE227397 | *Lu et al. (2023)* |
| *Synechocystis sp.* PCC 6803 | GSE132275 | Unpublished |
| *Xanthomonas albilineans* | GSE229478 | *Wimmer et al. (2024)* |
| *Xanthomonas oryzae* pv. oryzae KACC 10331 | GSE89651 | *Park et al. (2022)* |

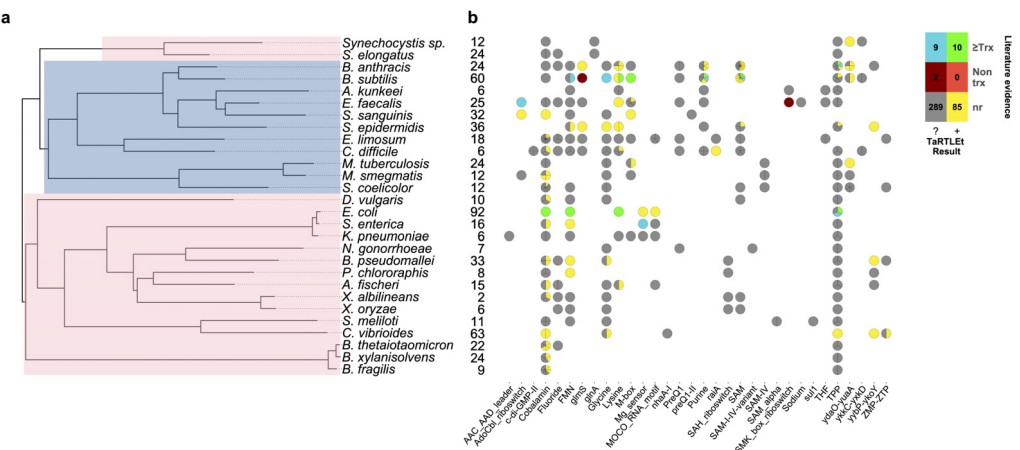

**Figure 6** **Compilation of `TaRTLEt` results for a collection of commonly studied bacteria, using RNA-seq data obtained from public GEO repositories.** (A) Phylogenetic relationships between bacteria studied with `TaRTLEt`. The tree is based on GTDB (*Parks et al., 2022*; *Rinke et al., 2021*; *Parks et al., 2020*; *Parks et al., 2018*) bacterial phylogenies (release 214.1) and built using the `ggtree` R package (*Yu et al., 2017*). Clades highlighted blue and pink represent Gram-positive and Gram-negative taxa respectively. (B) Evidence for condition-dependent transcription termination for every riboswitch representative, as found in the literature and by `TaRTLEt`. Counts at left give the number of experimental conditions for which paired-end RNA-seq data was available; a wider range of conditions increases the likelihood that the dataset will capture condition-dependent transcription termination. At each locus, evidence reported in the literature may support regulation by transcription termination, potentially among other mechanisms ("≥Trx"), or by non-transcription-termination mechanisms only ("Non trx"), or the locus may not have been characterized *in vitro* ("nr"); and `TaRTLEt` may be inconclusive ("?") or provide positive evidence ("+") for regulation by transcription termination. Where a species encodes multiple riboswitches of a given family, plotting symbols are divided into wedges for the different loci. Counts in the color key are tallies of riboswitch loci in each group.

here, leaving 366 loci tractable for further investigation. Second, `TaRTLEt` might detect no candidate transcription-termination events in the search space at a riboswitch locus (132 of 366 tractable loci); this could happen either because there is no terminator present in the region of interest, or because under the conditions tested the terminator is always inactive. In our validation set, this group included two loci previously characterized as acting *via* transcription termination, the *B. subtilis metI* SAM riboswitch (Fig. S4) and the *E. faecalis pduQ* AdoCbl riboswitch (Fig. S6) (*Winkler et al., 2003*; *DebRoy et al., 2014*).

Third, a locus with transcription-termination peaks might have no clusters that pass the mean fold coverage drop test, if the conditions examined yield consistently high rates of transcriptional read-through (whether because the conditions used sample only the ON state or because the riboswitch acts downstream of transcription). In our validation dataset, 20 of the 234 loci with passing peaks failed this test. This group includes two riboswitch loci (the *E. coli thiC* TPP and *B. subtilis pbuE* purine riboswitches) that have previously been shown to modulate transcription (*Winkler, Nahvi & Breaker, 2002*; *Chauvier et al., 2017*; *Mandal & Breaker, 2004*) but show no mean fold coverage drop here, suggesting that the range of conditions examined was too narrow to capture the OFF state for these riboswitches (Fig. 5). Interestingly, the purine ligand does vary enough across the

conditions tested in *B. subtilis* to cause a detectable switch in transcription termination at the *xpt* purine riboswitch (Fig. 6B, Fig. S4), suggesting that these two *B. subtilis* purine riboswitches are responsive to different ligand concentrations.

Finally, among the clusters at a locus that both include transcription-termination peaks and pass the mean test, none may pass the variance test (119 of the remaining 214 loci). A cluster might fail the variance test either if the variance of other peak clusters is large (*e.g.*, due to shallow coverage) or if the variance of the candidate cluster is small (due to a set of conditions that capture only one riboswitch regulatory state). In our validation set, five loci previously found to mediate transcription termination fell into this group: the *B. anthracis tenA* TPP (Fig. S5), *B. subtilis ribD* FMN, *gcvT* glycine and *yoaD* SAM (Fig. S4), and *S. enterica mgtA* $Mg^{2+}$ sensor riboswitches (Fig. S7) (*Welz & Breaker, 2007*; *Wickiser et al., 2005*; *Mandal et al., 2004*; *Winkler et al., 2003*; *Cromie et al., 2006*). Two of these, the *B. subtilis ribD* FMN riboswitch (Fig. S4, panel FMN|NC|2431617-) and the *S. enterica mgtA* $Mg^{2+}$ sensor riboswitch (Fig. S7, panel Mg_sensor|4699497+), appear to fail the variance test by being consistently OFF under the conditions tested; two more (*B. subtilis gcvT* glycine and *yoaD* SAM; Fig. S4, panels Glycine|NC|2549606- and SAM|NC|2025251-) are clearly consistent with conditional transcription termination but fall just shy of significance. Thus while our mean- and variance-based calling approach can in general be applied to existing datasets, the noisiness and the scope of the datasets examined can limit robust inference. However, where regulatory mechanism is already established by other means, the signal TaRTLEt exploits may still show riboswitch regulatory state even when the conditions sampled capture only ON or only OFF.

## TaRTLEt gives new insights into undercharacterized riboswitches

Thiamine pyrophosphate (TPP) riboswitches are the most abundant riboswitches in genome databases (*McCown et al., 2017*) and were identified in nearly all of the genomes in our validation set. *In vitro* characterization of this class has variously pointed to regulation at the level of transcription termination, translation initiation, or even both (*Sedlyarova et al., 2016*; *Winkler, Nahvi & Breaker, 2002*; *Chauvier et al., 2017*). In one notable case, *B. anthracis* encodes two TPP riboswitches in tandem upstream of the *tenA* gene, and each has been shown *in vitro* to be able to regulate transcription termination independently (*Welz & Breaker, 2007*). Of the 57 TPP riboswitches in the validation dataset, only six (two from *B. anthracis* and one each from *B. subtilis*, *S. epidermidis*, *E. coli*, and *C. vibrioides*) TPP representatives show transcriptional activity under the conditions tested. Intriguingly, the peak plot for *B. anthracis* showed transcriptional activity by the upstream but not the downstream *tenA* TPP riboswitch (Fig. S5, panels TPP|752099+ and TPP|752272+), suggesting that the two tandem riboswitches are responsive under different conditions *in vivo*.

A second very widespread riboswitch class binds cobalamin, with at least one example in all but four genomes in our validation dataset. Previous research has predicted that cobalamin riboswitches in Gram-positive taxa will typically work at the transcriptional level, while those in Gram-negative taxa will regulate translation (*Lennon & Batey, 2022*). Contra this expectation, we see clear evidence of transcriptionally-active cobalamin

riboswitches in the majority of Gram-negative taxa we investigated (Fig. 6; 15 Gram-negative taxa with cobalamin riboswitch loci, of which ten have at least one positive TaRTLEt result). Interestingly, however, in nearly every case, these taxa also contained one or more additional cobalamin riboswitches that showed no evidence of transcriptional regulation. These other riboswitches could either regulate other aspects of mRNA production, stability, or translation, or regulate transcription termination at a different range of ligand concentrations than this dataset captures. While TaRTLEt cannot distinguish between these possibilities with the data used here, the observation underscores the potential variability of riboswitches within a ligand class and highlights the utility of high-throughput analyses for identifying interesting candidates for experimental characterization.

TaRTLEt also offers insights into the less-studied riboswitch classes. For example, the molybdenum cofactor (MOCO) riboswitch in *E. coli* regulating the *moaA* operon is well-characterized structurally (*Amadei et al., 2023*) but not functionally. Our results indicate that this MOCO riboswitch modulates transcription termination. Across the 32 riboswitch families represented in our validation dataset, we were able to identify previous *in vitro* characterization for members of only 12 families, leaving 20 families completely uncharacterized. We find clear evidence for riboswitch-mediated transcription termination at 12 loci across five of these understudied families (MOCO, *raiA*, *ydaO-yuaA*, *yybP-ykoY*, and ZMP-ZTP families; Fig. 6). That is, TaRTLEt-enabled opportunistic analysis of data originally gathered for other purposes expands by nearly 50% the range of riboswitch families for which we have data-driven insights into regulatory mechanism.

## DISCUSSION

TaRTLEt's built-in visualizations facilitate user assessment of positive-call reliability. Clusters of peaks at multiple loci under multiple conditions can be rapidly inspected in peak plots, with the underlying coverage plots available to check results at lower degrees of abstraction. Thus, TaRTLEt serves as a much-needed high-throughput tool for preliminary assessment of riboswitch mechanism and can readily identify strong candidates for *in vitro* validation.

As currently implemented, TaRTLEt can analyze paired-end RNA-seq data to detect positive evidence of condition-dependent changes in transcription termination at riboswitch loci. The absence of such evidence cannot be used to infer a different mechanism of riboswitch action, because TaRTLEt's approach depends on having a dataset that captures both ON and OFF states. Thus, among the eight species in our validation dataset for which the RNA-seq collection included $\leq 9$ conditions, TaRTLEt could detect evidence for transcriptional regulation at any riboswitch in only four (Fig. 6B; compare to positive results in 19 of 21 species with data on $\geq 10$ conditions). Importantly, our approach supports inference from RNA-seq data collected across multiple different experiments, such that marginal additional data collection for the less well-studied species could make detection rates rise substantially.

## CONCLUSIONS

We identified a transcriptomic signal that should arise from riboswitch-mediated conditional regulation of transcription termination, developed `TaRTLEt` to apply convolution methods to extract the signal from existing public RNA-seq data, and demonstrated that this tool supports confident identification of riboswitches that control transcription termination. Our results with this method are in good agreement with the limited number of *in vitro* characterizations available, offer new insights into previously uncharacterized riboswitches, and emphasize that riboswitches within the same ligand class, even within the same organism, can exhibit different ligand sensitivities and act by different mechanisms. Whereas detailed *in vitro* characterization can establish which of several regulatory modes a single riboswitch uses, our high-throughput approach can ask which of any number of riboswitch loci appear to act by transcription termination and broadly survey the conditions under which this activity is invoked *in vivo*. Application of `TaRTLEt` to microbial transcriptomic and metatranscriptomic datasets therefore offers a path forward to understanding the regulation of a wide range of biosynthetic and transport pathways in biological contexts ranging from pure culture to natural communities *in situ*.

## ACKNOWLEDGEMENTS

This work is a contribution of the EMERGE Biology Integration Institute. We thank the members of EMERGE-BII, the VirSoil collaboration, and the Bagby lab for helpful discussions. This work used an allocation of computing time from the Ohio Supercomputer Center.

### Funding

Sarah C. Bagby and Sachit Kshatriya were supported by the Department of Energy Office of Science award number DE-SC0023307 and by the NSF Biology Integration Institutes Program award number 2022070. There was no additional external funding received for this study. The funders had no role in study design, data collection and analysis, decision to publish, or preparation of the manuscript.

### Grant Disclosures

The following grant information was disclosed by the authors:
The Department of Energy Office of Science: DE-SC0023307.
The NSF Biology Integration Institutes Program: 2022070.

### Competing Interests

The authors declare there are no competing interests.

### Author Contributions

- Sachit Kshatriya conceived and designed the experiments, performed the experiments, analyzed the data, prepared figures and/or tables, authored or reviewed drafts of the article, and approved the final draft.

- Sarah C. Bagby conceived and designed the experiments, analyzed the data, prepared figures and/or tables, authored or reviewed drafts of the article, secured funding, and approved the final draft.

## Data Availability

All validation data is public; validation data harvesting, processing, and analysis code is available at Github and Zenodo:

- https://github.com/lepton-7/tartlet-pub,
- Sachit Kshatriya. (2025). lepton-7/tartlet: v1.0.0: Initial Release (v1.0.0). Zenodo. https://doi.org/10.5281/zenodo.14866620
- Sachit Kshatriya, & Sarah C. Bagby. (2025). lepton-7/tartlet-pub: DOI Release (v1.1). Zenodo. https://doi.org/10.5281/zenodo.14867191.

## Supplemental Information

Supplemental information for this article can be found online at http://dx.doi.org/10.7717/peerj.19418#supplemental-information.

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
