# Peer review of "TaRTLEt: Transcriptionally-active Riboswitch Tracer Leveraging Edge deTection"

_PeerJ, doi:10.7717/peerj.19418_

## Round 0.1 · original submission · Major Revisions

Your manuscript has been reviewed by three experts, and while they acknowledge the potential value of your work, all generally recommend major revisions. Their key concerns include:

1. Definition of algorithmic purpose and advantages (Reviewer 1):
The specific objectives and advantages of the algorithm need to be more clearly stated.
2. Clarification of output interpretation (Reviewer 3, Reviewer 2):
The current explanation of TaRTLEt’s output is insufficient, making it difficult for users to interpret the results. Moreover, the method's limitations need to be highlighted more (Reviewer 2).
3. Pipeline Walkthrough (Reviewer 1, Reviewer 2):
The lack of a detailed, step-by-step description of your pipeline hinders readers’ ability to fully understand and trust the results.
4. Content of current Figure 1 (Reviewer 1), Figure 3 (Reviewer 1), and Figure 6 (Reviewer 2).

In addition, there are valuable suggestions that, while optional for implementation, need a thoughtful response in your revision:
- Introduce a Confidence Measure (Reviewer 3):
Consider developing a confidence metric for multiple hits based on input data quality and completeness.
- Broaden Application Scope (Reviewer 3):
Discuss and consider testing potential applications of your method to related fields, such as detecting splicing-controlling riboswitches.
- Clarify Regulatory Mechanisms (Reviewer 1):
Expand on which regulatory mechanisms your method can and cannot detect to provide a clearer context for its capabilities.

All three reviewers have also emphasized the need to improve the manuscript’s overall phrasing and readability. Please revise the text to address these concerns and enhance its clarity and accessibility.

We look forward to receiving your revised submission along with a detailed point-by-point response to the reviewers' comments.

Reviewer 1 ·

Basic reporting

This article is devoted to the accurate mapping of the beginning and end of the RNA sequence from RNA-seq data and the application of this method to the search for riboswitches. It is assumed that we have RNA-seq data in different conditions and in one of the conditions riboswitch is activated and premature transcription termination occurs. The proposed pipeline allows us to accurately map the start and end of riboswitches based on sequencing. Here we should keep in mind several points not noted in the paper. Firstly, riboswitch is not the only mechanism of regulation based on premature transcription termination. Transcriptional attenuation is a classic example of such regulation. In addition, T-box regulatory systems operate by a similar mechanism of premature termination. These systems, unlike riboswitch, do not respond directly to the presence of ligand, but indirectly. Second, the software finds not only riboswitches (and other attenuators), but also the signal of transcription termination. Third, riboswitches not only prematurely terminate transcription but also silence translation by hiding translation initiation sites. This type of regulation cannot be detected by the proposed methods.

The purpose of the work is not clearly defined. If the programme searches for known riboswitches? There are algorithms that do this. What are the advantages of the proposed approach? What is input and output of the pipeline.

Experimental design

Although the ideas underlying the work seem to be quite simple, the text is very difficult to comprehend. The algorithm should be presented in a much simpler way. For example, Figure 1 is not very informative, and peak calling procedure and clustering are not supported by figures. Seems pipeline workflow should be given

Validity of the findings

This is a methodological paper. No new significant findings are presented.

Additional comments

Some specific comments:
1. Although the ideas underlying the work seem to be quite simple, the text is very difficult to comprehend. The algorithm should be presented in a much simpler way. For example, Figure 1 is not very informative, and peak calling procedure and clustering are not supported by figures.
2. Figure 3 - What do lines and line colours mean? Figures like this appear in the results section. It seems to be an important figure. The graph should be described more clearly - the user should understand the graph.
3. It would be good to give a schematic of the pipeline. There is a section on Riboswitch reference generation in the text. Does it mean that the pipeline is oriented to search for known riboswitches? Or is the presented pipeline looking for all possible cases of attenuation? If the former, it is not clear why the exact mapping of the ends is necessary? Isn't it enough to apply known methods to search for riboswitches and see if they are major gene expression changes?
4. Line 236: “...density function with sigma = 1.5”. Is this value measured in nucleotides? If so, 51 positions are too precise.

The text should be rewritten in a clearer way with appropriate illustrations.

Reviewer 2 ·

Basic reporting

The manuscript is very clearly written. There are abundant appropriate
references. The structure is professional with relevant and sufficient
figures and tables, and the data is available on github.

Experimental design

Kshatriya and Bagby present a novel idea and method for investigating
the mechanism of action of predicted riboswitches based on paired-end
RNAseq data. The authors do an excellent job of briefly introducing
riboswitches and the various ways they function. They present a clever
and novel idea for addressing the question of whether individual
riboswitches act by transcription termination, and offer a tool,
TaRTLEt, that implements that idea based on analysis of paired end
RNA-seq datasets. The methods behind their TaRTLEt tool are clearly
explained and the tool itself is publicly available. The authors
present and discuss the results of using the tool on publicly
available RNA seq datasets coupled with riboswitch predictions made by
a homology search tool. Their results agree with some previously
published experimental results and offer new insights for some
riboswitch predictions.

The methods are described in detail. Regarding reproducibility,
Page 5, line 17 reads "A complete walk-through of TaRTLEt usage
and the analysis of data generated by TaRTLEt is available at the
tool GitHub repository (https://github.com/lepton-7/tartlet).", but it
was unclear how to access the 'walk-through'. I also looked at
"https://github.com/lepton-7/tartlet-pub" thinking perhaps it was
there but also couldn't find any obvious walk-through. Could the
authors either clarify what is meant by 'walk-through', or add an
obvious walk-through to the github repo?

For the walk-through, my opinion is that it would be most helpful if
there was a tutorial or guide that showed each of the steps required
to perform the entire TaRTLEt based analysis, including Infernal,
Prodigal, and HISAT2 preliminary steps for at least one genome and
read dataset from the validation dataset, if this is feasible. This
would allow a user to perform all the steps on a subset of the
validation dataset for which the expected output (included as part of
the walkthrough) is known. By checking the output they get versus the
expected output they would be able to determine if they were
performing the steps correctly. This would give them confidence to do
analyses of other, perhaps their own, datasets.

Validity of the findings

The conclusions are clearly presented and are relevant and justified
given their results.

Additional comments

Minor additional comments:

1. TaRTLEt relies on riboswitch predictions in genome sequences, as
well as paired-end sequence datasets. It does not allow the
prediction of novel riboswitches, since a riboswitch must be known
before an Infernal model can be built for it. I think it would help
some readers if the authors make this more clear early in the
paper. Page 2, line 69-71 reads:

"Here, we describe the new tool TaRTLEt (Transcriptionally-active
Riboswitch Tracer Leveraging Edge deTection), which implements a
method to identify riboswitches that operate via alteration of
transcription termination efficiency using (a) high-throughput
computational approaches and (b) publicly available RNA-seq data."

My suggestion is to modify the above sentence, or add text
somewhere near it, to indicate that TaRTLEt can only identify such
riboswitches from a list of candidate riboswitches, which
themselves are defined as candidates based on homology search
tools. That is, it cannot identify novel riboswitches (from
previously unknown families) based solely on publicly available
RNA-seq data.

2. Figure 6 is a very informative figure, but I have a few
suggestions/questions:

* please indicate which bacteria are gram positive versus gram
negative on the tree or elsewhere. This is relevant to your point
about Cobalamin being transcriptionally active in more than half
of the gram negative bacteria you have data for.

* you wrote " Of the 57 TPP riboswitches in the validation dataset,
only 3 (1 from E. coli and 2 from B. anthracis) TPP
representatives show transcriptional activity under the
conditions tested." But it looks to me like there are 2 yellow
and 1 green in B. anthracis - wouldn't that make 3 TPP examples
in B. anthracis that have a positive TaRTLEt result (and show
transcriptional activity).

* please clarify what the numbers ranging from 2 to 92 directly
under the 'b' mean. I believe it is number of conditions tested
in the RNA-seq datasets, but only was able to guess this after
reading the Discussion.

* in the text discussing figure 6, you wrote (page 11, "Previous
research indicates that cobalamin riboswitches in Gram-positive
taxa typically work at the transcriptional level, while those in
Gram-negative bacteria regulate translation". Doesn't this
suggest that at least some of the Gram-positive bacteria should
have blue wedges instead of grey, since they are ? for TaRTLEt
but the literature suggests they work at the transcriptional
level?

3. At the beginning of page 12, you list the number of loci and
species for which TaRTLEt results indicate "condition-dependent
modulation of termination efficiency". I think it would be
informative to the reader if you were able to include in these
counts the number that agreed vs were contrary to literature
evidence, as indicated in Figure 6. This would summarize the
findings with respect to their agreement with the literature. This
information could alternatively be included somehow in Figure 6
(sums of number of wedges of each color, for example).

4. Page 7, line 231-232 reads "The coverage threshold can be adjusted
with the --min-cov-depth option (default 50; the validation presented
here uses 15)". Why did you not use the default for the validation
presented in the paper? Or change the default to 15? I think many
readers will be curious about this and it would be helpful if you
can address this question in the text.

Typo:
Figure 3 caption, sentence 1 reads "Evidence of conditional
transcription termination at a riboswitch 3'.". I think you omitted a
word at the end of the sentence.

·

Basic reporting

The article meets most of the basic reporting standards; however, some aspects require clarification and elaboration, see below

Experimental design

The experimental design is generally sound and within the scope of the journal, but certain details need further clarification to ensure transparency and reproducibility

1. What are negative controls? I can imagine 2 negative controls: (1) non-riboswitch, and (2) riboswitch acting through a different mechanism. The authors mention the second group:

Conversely, the B. subtilis glmS riboswitch, 310 the Enterococcus faecalis SMK riboswitch, and the E. coli lysC lysine riboswitch have been shown to 311 act via other mechanisms (Collins et al., 2007; Fuchs et al., 2007; Caron et al., 2012), and for these loci 312 TaRTLEt finds no evidence of riboswitch regulation by transcription termination across the conditions 313 represented in the dataset.

SInce authors envision this as a tool for scientific community to use, it is critical understand the nature and frequency of false positives and false negatives. However, there is no quantitative analysis to compare for the FP and FN predictions. The authors do talk about the nature of false negatives, but not false positives.

2. I don’t think the paper describes sufficiently how the user should be interpreting the outputs of TaRTLEt. THe authors write:
TaRTLEt outputs data on two layers of abstraction: locus plots of fragment coverage with the convolution output for each riboswitch locus and RNA-seq treatment, and the summary peak log.
They also write:
As currently implemented, TaRTLEt can analyze paired-end RNA-seq data to detect positive evidence of condition-dependent changes in transcription termination at riboswitch loci.
And in the figure 6, they demonstrate “Evidence for condition-dependent termination efficacy for every riboswitch representative, as found in the literature and by TaRTLEt.”. However, it is not entirely clear to me how this evidence is extracted from TaRTLEt’s output. The authors use the word “Mechanism” repeatedly; yet it is not clear what criteria they have used to identify the riboswitches acting through one or another mechanism.

3. Additionally, I don’t quite understand what happens when several cuts are predicted by TaRTLEt: is there a probability assigned to each of them? Is the working hypothesis that one of the cut sites is correct, while the others are not? It would be nice to expand on that, especially in the context of Fig. 5

Validity of the findings

4. It is possible that the input data differs for different bacterial genomes (say more RNAseq data available in one case than the other); therefore, the tool’s predictions will have different level of confidence. Is there a way to formalize it and to report the level of confidence?

Additional comments

5. Is it possible to apply the same tool to detection of ribozymes, or splicing-controlling riboswitches mentioned by the authors?

6. What exactly do authors mean when setting up the problem as “Yet our understanding of riboswitch function in vivo remains patchy.” and “the full range of riboswitch regulatory mechanisms remains unknown”? If this refers to lack of understanding of possible mechanisms or the ligands, the current work does not address these problems. I believe the introduction needs to set up the problem the tool is trying to solve more specifically.

---

## Round 0.2 · Minor Revisions

Please, consider the remaining minor comments from the reviewers. Based on them, the addition of the schematics of the method's workflow is essential.

Reviewer 1 ·

Basic reporting

The paper is clear written. The proposed approach allows to find the transcription attenuation effect, in particular, the riboswitches. The updated version of the paper is aceptible.

Experimental design

The algorithm is described in suffisient details, but the workflow chart should be presented.

Validity of the findings

The manuscript not only defends the successes, but also describes the limitations of the approach. The observed biological effects are discussed.

Additional comments

The workflow chart for the method should be presented.

Reviewer 2 ·

Basic reporting

The manuscript has been substantially improved since the first version I read.
(As I wrote in the initial review):
The manuscript is very clearly written. There are abundant appropriate
references. The structure is professional with relevant and sufficient
figures and tables, and the data is available on github.

Experimental design

(As I wrote in the initial review):
Kshatriya and Bagby present a novel idea and method for investigating
the mechanism of action of predicted riboswitches based on paired-end
RNAseq data. The authors do an excellent job of briefly introducing
riboswitches and the various ways they function. They present a clever
and novel idea for addressing the question of whether individual
riboswitches act by transcription termination, and offer a tool,
TaRTLEt, that implements that idea based on analysis of paired end
RNA-seq datasets. The methods behind their TaRTLEt tool are clearly
explained and the tool itself is publicly available. The authors
present and discuss the results of using the tool on publicly
available RNA seq datasets coupled with riboswitch predictions made by
a homology search tool. Their results agree with some previously
published experimental results and offer new insights for some
riboswitch predictions.

The authors updated the github page to include the walkthrough, which wasn't present when I reviewed the initial version. I have reviewed the walkthrough and it is comprehensive and clear.

Validity of the findings

(As I wrote in the initial review):
The conclusions are clearly presented and are relevant and justified
given their results.

Additional comments

In my initial review, I had several minor comments. All of these were addressed to my satisfaction by the authors in the new version of the manuscript.

---

## Round 0.3 · accepted · Accept

As an Editor of PeerJ, I can conclude that your new Supplementary Figure now addresses the concerns of Reviewer 1 and significantly benefits the manuscript.

At the stage of proof-reading the manuscript, please, pay attention to the presence of all supplementary material, as well as proper references to it in the main text.